# Predicting Urban Waterlogging Risks by Regression Models and Internet Open-Data Sources

**Ducthien Tran** [1,2], **Dawei Xu** [1,*], **Vanha Dang** [2] **and Abdulfattah.A.Q. Alwah** [1,3]

[1]  College of Landscape Architecture, Northeast Forestry University, Harbin 150040, China;
    thientranduc.udt@gmail.com (D.T.); fattah_alwah@ibbuniv.edu.ye (A.A.Q.A.)

[2]  College of Landscape Architecture and Urban Greenning, Vietnam National University of Forestry,
    Hanoi 156200, Viet Nam; halndt2010@gmail.com

[3]  Faculty of Engineering and Architecture, IBB University, Ibb 00000, Yemen

*  Correspondence: xdw.ysm@gmail.com; Tel.: +86-133-0463-7599

**Abstract:** In the context of climate change and rapid urbanization, urban waterlogging risks due to rainstorms are becoming more frequent and serious in developing countries. One of the most important means of solving this problem lies in elucidating the roles played by the spatial factors of urban surfaces that cause urban waterlogging, as well as in predicting urban waterlogging risks. We applied a regression model in ArcGIS with internet open-data sources to predict the probabilities of urban waterlogging risks in Hanoi, Vietnam, during the period 2012–2018 by considering six spatial factors of urban surfaces: population density (POP-Dens), road density (Road-Dens), distances from water bodies (DW-Dist), impervious surface percentage (ISP), normalized difference vegetation index (NDVI), and digital elevation model (DEM). The results show that the frequency of urban waterlogging occurrences is positively related to the first four factors but negatively related to NDVI, and DEM is not an important explanatory factor in the study area. The model achieved a good modeling effect and was able to explain the urban waterlogging risk with a confidence level of 67.6%. These results represent an important analytic step for urban development strategic planners in optimizing the spatial factors of urban surfaces to prevent and control urban waterlogging.

**Keywords:** urban waterlogging risk; regression models; internet open-data sources; spatial analysis; ArcGIS

## 1. Introduction

Rapid urbanization is accelerating the transformation of urban natural surfaces, usually by reducing ecological land and increasing the impervious cover areas, which prevents rainwater from penetrating the soil [1–8]. Urban streets act as streams by collecting rainwater, thereby increasing the volume of surface runoff, which is one of the most common causes of urban waterlogging [9–11]. In the context of climate change, waterlogging due to rainstorms is becoming more frequent and serious in developing countries and is causing losses to the economy, as well as affecting the living environment, people's daily life, and the sustainable development strategies of cities [12]. In 2017, Vietnam's urbanization exceeded 35% [13] and it is estimated to reach nearly 56% by 2050 [14]. As a center of national politics in Vietnam, Hanoi is an important and typical city in the Red River Delta, as well as the most representative city of serious urban waterlogging [15–18]. Due to rapid urbanization, the permeable areas have been replaced by hard surfaces. Such swift rural–urban transformation [19–21] has had adverse impacts on the complex environmental water management system of dikes, pumping stations, and sluices [17,18,22].

Compounding the negative influence of this development is the fact that the city's pump stations are operating beyond their capacities, in addition to Hanoi's outdated underground drainage system being rendered inoperative when rainfall is higher than 100 mm/h [17]. On 30 October 2008, Hanoi experienced the most extreme inner-city floods to date. The death toll reached a devastating total of 22 people. Many streets were flooded, with rainfall of up to 450 mm, and some actually collapsed. Nearly 35,000 households were flooded. The estimated damage exceeded 3000 billion VND (1.6 million USD) [23]. As a result, both preventing and controlling the hidden dangers resulting from urban waterlogging have become an urgent mission for Hanoi's municipal governments.

The current lack of a field investigation database and appropriate assessment tools has impeded the conduct of adequate studies to analyze and quantify inundations as well as to identify the relationship between the spatial factors of urban surfaces and the waterlogging frequency in the inner city of Hanoi. To overcome this problem, the idea of a regression model in ArcGIS utilizing internet open-data sources has been proposed to elucidate the roles played by the spatial factors of urban surfaces that cause urban waterlogging and to predict urban waterlogging risks.

Current research on urban waterlogging problems includes the causes of urban waterlogging [24–28], urban stormwater control [29,30], urban stormwater management [31–36], urban waterlogging simulation [37–39], and the spatial assessment of urban waterlogging risks and vulnerability in flood-prone areas [12,40–43]. Some assessment models for urban waterlogging risk have been proposed. In general, previous studies have focused on GIS-based spatial analysis, including simulation models and scenario simulations that use remote sensing datasets, terrain data, rainfall data, meteorology data, hydrological data, socioeconomic data, urban drainage systems, etc. [28,39,44–49]. In addition, hydrologic- and hydrodynamic-based mathematical simulation models are among the main methods for evaluating urban waterlogging risks [12,37,40,41,50,51].

In mathematical statistics, regression analysis is one of the most widely used methods for exploring and exploiting the relationships between dependent and explanatory variables [52], i.e., regression analysis explains the changes in the dependent variable when one of the explanatory variables is varied while the other explanatory variables are fixed. Regression analysis is widely used in epidemiology [53,54], economics [55,56], environmental science [57,58], behavioral and social sciences [59,60], etc. In the spatial statistics of ArcGIS, regression analysis is used as a spatial regression technique, which is increasingly applied in geography and other disciplines. When applied properly, a model's analytical results provide reliable and accurate statistics to examine and estimate linear relationships [61]. Efforts to apply the regression model in ArcGIS for estimating urban waterlogging risks have been made, particularly in the last year. For example, a geographically weighted regression model was developed to assess the spatiotemporal variance in the impact of impervious surface expansion on urban rainstorm waterlogging in Guangzhou, China, during 2012–2018 [26]. Wang et al. (2017) applied a geographically weighted regression model with a database of waterlogging events to examine the spatially explicit relationships between waterlogging frequency and the spatial explanatory factors of urban pluvial floods in Shanghai, China, during 1997–2013 [62].

Open data are understood as different types of data that can be provided freely to everyone for development, processing, storage, and organization according to the specific needs of users and without restrictions from copyrights, patents, or other mechanisms of control [63]. Currently, open urban data are very complete, easy to access, and highly quantitative. As a vital complement to traditional investigative data, open urban data can contribute to urban management and the solving of urban problems [64]. In urban research, many scholars acquire urban data by various means, such as through satellite map data, topographic data, population data, traffic data, telecom data, economy data, and society data by utilizing search engines [65–70]. For example, Hollenstein and Purves (2010 explored the urban spatial structure of London and Chicago metropolitan areas with locational information from Flickr [71]. Lüscher and Weibel (2013) realized the division of urban central areas in the UK by distinguishing the urban spatial geographical features by the terrain map database [72–74]. Wang et al. (2014) and Niu et al. (2014) explored and analyzed urban spatial structure using mobile

signal data [75,76]. Liu et al. (2014) studied the relationship between geographic location and human behavior based on location information [77].

Yu and Ai (2015) used traffic data sources to analyze and visualize traffic density distribution using the Kernel density method in ArcMap to highlight the advantages of the expression methods of the Venn diagram. The results of this study provide technical support for urban planning in the new era [78]. Lin et al. (2018) analyzed and assessed the urban waterlogging risk of each district in China's Xiamen city by considering both urban internet open data and field investigation. The result indicates that internet open-data sources have great potential for urban waterlogging risk assessment [79]. In addition, the wide application of satellite-based precipitation data has greatly promoted hydro-meteorological research in areas where precipitation observations are scarce [80–86]. Satellite precipitation estimates are playing an increasingly important role in providing useful precipitation information for the optimal management of water resources, flood mitigation, drought warning, etc. [81,87,88]. However, the various types of satellite-derived precipitation data used here are only for regional studies.

From the above literature reviews, we found that urban waterlogging has been studied by a variety of methods that require large amounts of data, including field investigation data, which are generally difficult to obtain. In addition, in the current era of open urban data, regression modeling has provided a powerful method for discovering knowledge and inferring probabilities under conditions of uncertainty and has been widely applied to estimating the likelihood of various hazards. Regression modeling should be considered to describe possible relationships among spatial factors, which would probably improve the prediction efficiencies of urban waterlogging risk assessments. Finally, the integration of GIS technologies is one of the most inevitable trends in research on urban waterlogging.

Taking Hanoi, Vietnam as a case study, we aimed to analyze the spatial characteristics of waterlogging areas and to elucidate the spatial factors of urban surfaces as a cause of urban waterlogging. At the same time, we applied a regression model in ArcGIS with internet open-data sources to predict the possibilities of urban waterlogging risks in Hanoi for the period of 2012–2018.

## 2. Study Area and Data Sources

### 2.1. Study Area

Hanoi is the capital of the Socialist Republic of Vietnam and is the country's second largest city, as well as the center of culture, science, education, economics, and international trade. With the advantages of a strategic geopolitical location and a long-standing development history, Hanoi also plays a key role in the development of the essential economic zone in every aspect—not only in North Vietnam, but also in the whole country. Hanoi is located at the center of the Red River Delta (Figure 1), which belongs to a tropical region that is greatly affected by monsoons. The climate in Hanoi is very comfortable and four common seasons are discernible: January is the coldest month, with a monthly mean temperature of 16.5 °C, and July is the hottest, at 29.5 °C. The annual average temperature is 23 °C and precipitation is 1760 mm, with about 114 rainy days/year. Every year, the rainy season is from May to October, with the highest rainfall in August (an average rainfall of 274 mm) and the lowest rainfall in January, February, and December (an average rainfall of 19 mm). The area receives about 1562 h of sunshine per year. The average altitude is 5–10 m above mean sea level with a terrain gradually sloping down in the North–South and West–East directions. Many areas in old Hanoi (including Ba Dinh, Dong Da, Hai Ba Trung, and Hoan Kiem districts) have an elevation of +5 m, which is significantly lower than the riverbed of the Red River (which has an average elevation of +7 to +8 m) [22]. As a result, Hanoi is very vulnerable to flooding during the rainy season [16]. In 2018, Hanoi had a population of 7.78 million and an area of 3358.9 km$^2$, including 12 districts, one provincial town, and 17 rural districts. The study area is located in the inner city of Hanoi, which has an incredibly dense population and spans 454.9 km$^2$, including 12 urban and two rural districts (Figure 1), as well as a rich lake and river network [16].

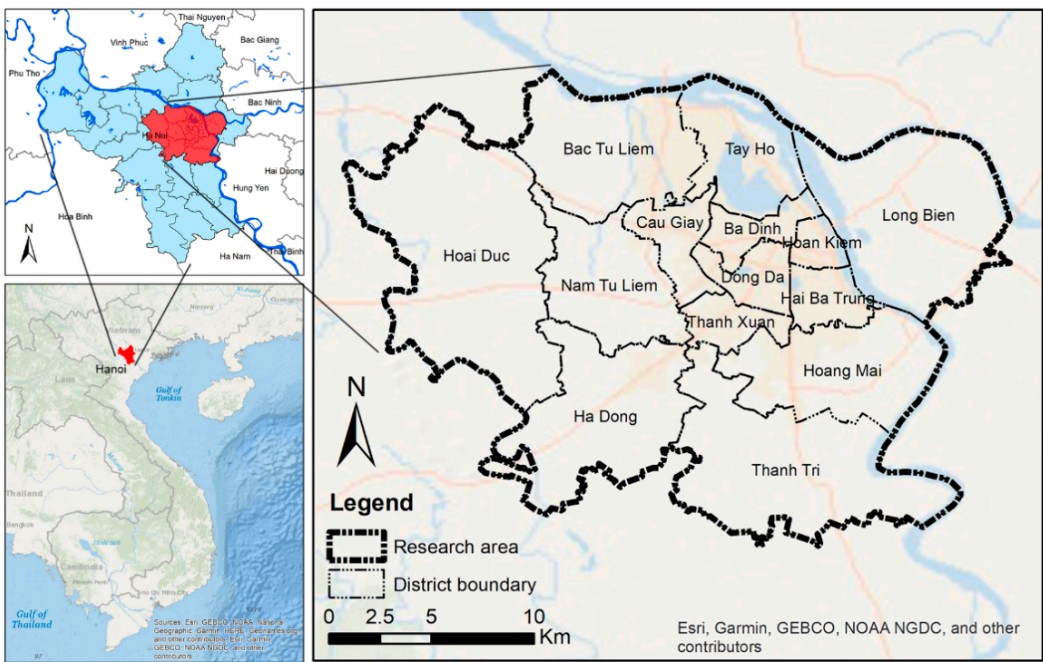

**Figure 1.** Location of the study area.

*2.2. Data Sources*

Acquiring data on the waterlogging spots is key to the evaluation of waterlogging risks. We used Google's search engine [89] to find news reports about urban waterlogging on the internet and selected the most useful reports. A total of 21 rainstorms were identified that had created a large waterlogged area in Hanoi during the period 2012–2018, and about 98 reports with 148 waterlogging spots were adopted for analysis (Figure 2). We used 30-m resolution Landsat 8 OLI/TIRS data by United States Geological Survey (USGS) from 7 June 2018 (path/row 127/45) as the remote sensing imagery for this study. The selected images had less than 10% cloud cover, which is representative of the season. A Digital Elevation Model (DEM) used 30-m resolution ASTER Global DEM V2 data by USGS from 17 October 2011. High-resolution remote sensing images of Hanoi in 2018 were taken from Google Earth's aerial photographs. Administrative areas were taken from DIVA-GIS, road maps were taken from OpenStreetMap, and population information was derived from data provided by Hanoi Portal and the Hanoi Urban Planning Institute. All data sources are shown in Table 1.

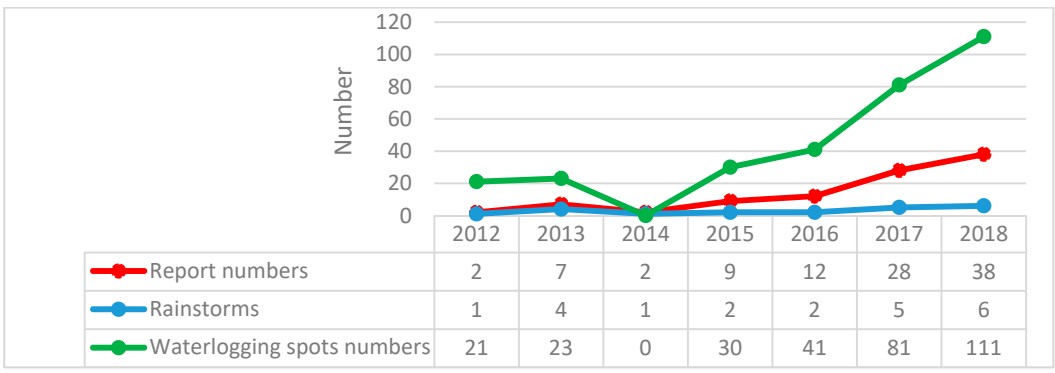

**Figure 2.** Report numbers of the period 2012–2018.

**Table 1.** List of data sources.

| Data | Format | Time | Source |
|---|---|---|---|
| Waterlogging information | Text | 2012–2018 | Google [89] |
| Landsat satellite data | Raster | 2018-6-07 | USGS [90] |
| Topographic data | Raster | 2011-10-17 | USGS [91] |
| High-resolution remote sensing image | Raster | 2018 | Google Earth Pro |
| Administrative areas | Shapefile | 2018 | DIVA-GIS [92] |
| Traffic map data | Shapefile | 2018 | OpenStreetMap [93] |
| Population data | Text | 2013–2017 | Hanoi Portal [94] Hanoi Urban Planning Institute [95] |

## 3. Methodology

### 3.1. Regression Model

A regression model was implemented in ArcGIS to model, examine, and explore spatial relationships in order to explain the factors behind the observed spatial patterns [61]. Ordinary Least Squares (OLS) Regression is a traditional regression model for quantifying the relationships between dependent and explanatory variables in both univariate and multivariate analyses [62] by providing a global model of the variables, but it is incapable of explaining the heterogeneity and non-stationarity of spatial relationships among geographical data [96,97]. Geographically Weighted Regression (GWR) is also a spatial regression technique, which provides a local model of the variables. GWR obeys Tobler's first law of geography, and fully considers the spatial location of adjacent factors [26]. GWR can remedy the disadvantage of OLS to explore the spatial non-stationarity of relationships among geographical data [98–100]. The formula for our regression model is (1)

$$Y = \beta_0 + \sum_{k=1}^{n} \beta_k X_k + \varepsilon \qquad (k = 1, 2, \ldots n) \tag{1}$$

where $Y$ is the dependent variable, which is what we are trying to model or predict; $X$ represents the explanatory variables, which, we believe, influence or help explain the dependent variable; $\beta$ refers to the coefficients, which are values that are computed by the regression tool to reflect the relationship of each explanatory variable and the strength of that relationship with the dependent variable; and $\varepsilon$ refers to the residuals, which are the portions of the dependent variable that are not explained by the model.

### 3.2. Data Processing

#### 3.2.1. Waterlogging Risk Areas

Waterlogging spots were considered to be the most important resource in urban waterlogging risk analysis. Information, including location, time, and occurrence frequency, about urban waterlogging spots was integrated with Microsoft Excel, then digitalized with ArcGIS to construct the spatial distributions of the spots, which were regarded as risk sources.

Areas that are 500 m away from the boundaries of each source were regarded as risk areas, because urban public service works in Vietnam within a residential unit (school, market, etc.) must have a service radius that does not exceed 500 m. The distance for pedestrians to travel from their places of residence or work to public car parks must also not exceed 500 m [101].

Therefore, we adopted 500 m as the urban waterlogging risk radius and applied the Buffer tool in ArcGIS to analyze the urban waterlogging risk areas.

### 3.2.2. Waterlogging Spot Density

We assumed that each district in the study area had experienced the same rainstorm, i.e., had been subjected to the same rainfall intensity and level of danger. Because the waterlogging events are recorded as points, it is very difficult to show a neighborhood's waterlogging situation. Therefore, we applied the Kernel Density Estimation (KDE) tool in ArcGIS to calculate the densities of geographical objects [102]. KDE can be calculated for point or line features. [103] described KDE as a method that can handle comprehensive estimates of the distribution of events based on a finite data pattern. The purpose of using KDE in this study was to create a smooth density surface of point or line events in space by calculating the event intensity as density estimates [104].

The general density estimation function is calculated as follows [105]

$$f(x) = \frac{1}{nh} \sum_{i=1}^{n} \frac{K(x - x_i)}{h} \qquad (i = 1, 2, \ldots, n) \qquad (2)$$

where $x_i$ stands for the value of the variable $x$ at location $i$, $n$ signifies the total number of locations, $h$ denotes the bandwidth or smoothing parameter, and $K$ represents the kernel function, as explained in an earlier report [103].

### 3.2.3. The Relationships of Waterlogging Spots

We applied the Getis-Ord Gi* statistic of the Hot Spot Analysis (HSA) tool in ArcGIS to evaluate the existence and the relationships of waterlogging spots. The results show which locations have many waterlogging spots and help predict the causes of such spots. HSA is a type of spatial autocorrelation analysis [106] that helps us to understand if an event would create spatial clusters, as well as the impact of that event on the surrounding events, i.e., the HSA tool works by looking at the relationships of each event within the context of a neighboring event. A high-value event may not be a statistically significant hot spot. In order for an event to become a statistically significant hot spot, the event must be of high value and must be surrounded by other events with high values as well [107]. This method is often used in research in social sciences, such as criminology [108] and epidemiology [109,110]. The indicator, Getis-Ord Gi*, is defined as follows (3, 4, 5) [107]

$$G_i^* = \frac{\sum_{j=1}^{n} \omega_{i,j} x_j - \overline{X} \sum_{j=1}^{n} \omega_{i,j}}{S \sqrt{\frac{\left[ n \sum_{j=1}^{n} \omega_{i,j}^2 - \left( \sum_{j=1}^{n} \omega_{i,j} \right)^2 \right]}{n-1}}} \qquad (j = 1, 2, \ldots, n) \qquad (3)$$

where $x_j$ is the attribute value for feature $j$; $\omega_{i,j}$ is the spatial weight between features $i$ and $j$; $n$ is equal to the total number of features; and

$$\overline{X} = \frac{\sum_{j=1}^{n} x_j}{n}; \qquad S = \sqrt{\frac{\sum_{j=1}^{n} x_j^2}{n} - \left( \overline{X} \right)^2} \qquad (j = 1, 2, \ldots, n). \qquad (4)$$

### 3.2.4. Spatial Autocorrelation of Waterlogging Spots

We applied the Spatial Autocorrelation (Moran's I) tool in ArcGIS to measure the spatial autocorrelation of waterlogging spots, which is the dependence of waterlogging spots on space that is expressed as clustered, dispersed, or random. The Moran's I is calculated as follows (5)

$$I_i = \frac{y_i - \overline{y}}{s^2} \sum_{j=1}^{n} w_{ij} (y_i - \overline{y}); \qquad (j = 1, 2, \ldots, n) \qquad (5)$$

where $I_i$ represents Moran's index of the research unit $i$, $s^2$ is the discrete variance of $y_i$, $\overline{y}$ is the mean value of $y$, and $w_{ij}$ is the element of the weight matrix $w$. The adjacent relation is of the Queen type, i.e., the grids are adjacent to each other as long as there is a common boundary or vertex [26,111].

### 3.2.5. Attribute Extraction

To extract the attribute values of the spatial factors, we applied the Zonal Statistics as Table (ZST) tool in ArcGIS to calculate statistics for the values of the spatial factor raster. In ZST analysis, a statistic is calculated for each zone to extract the attribute defined by this zone's dataset (waterlogging risk areas) according to the values from another raster (spatial factors). In addition, to calculate the distance characteristics from waterlogging spots to the rivers and lakes in Hanoi, we used the Extract Multi-Values to Points (EMVP) tool in ArcGIS to extract the cell values of the Input raster at locations specified in the Input point feature, and we recorded the values in the attribute table of the Input point feature.

### 3.3. Explanatory Variables

In general, urban waterlogging surpasses the capacity of the urban drainage system, and thus waterlogging disasters happen. Compounding this fact are the city's pump stations operating beyond their capacities and Hanoi's outdated underground drainage system being rendered inoperative when rainfall is higher than 100 mm/hour [17]. In addition, as the study area is 454.9 km$^2$, the variation in precipitation can be limited. Therefore, this study did not consider precipitation, pump stations, or underground drainage systems, but was instead focused only on six spatial factors, namely, population density (POP-Dens), road density (Road-Dens), distance from water bodies (DW-Dist), impervious surface percentage (ISP), normalized difference vegetation index (NDVI), digital elevation model (DEM), and impervious surface (IS) of urban surfaces, to explain the urban waterlogging risks (Figure 3).

Currently, many areas in the core of Hanoi's inner city have very high population densities, which are areas that can be considered as slums with green spaces and technical infrastructures that are lower than the national standards [112]. We checked the population density data and location information of waterlogging spots and found that most of the waterlogging spots that occurred with high frequency were located in areas with high population densities (Figure 3a). Therefore, we used population density as one of the explanatory variables in our regression model.

Roads are covered by nearly impervious materials such as asphalt, concrete, brick, or stone, which render an area prone to waterlogging when it has a high road density. With the data from Hanoi's OpenStreetMap road network in September 2018, the KDE tool was used to calculate the road density in the study area (Figure 3b).

Elevation is typically considered to be one of the most influential factors in flood inundation [113]. In this study, we used the digital elevation model (DEM) to study the impact of urban surface elevation on urban waterlogging risk (Figure 3c).

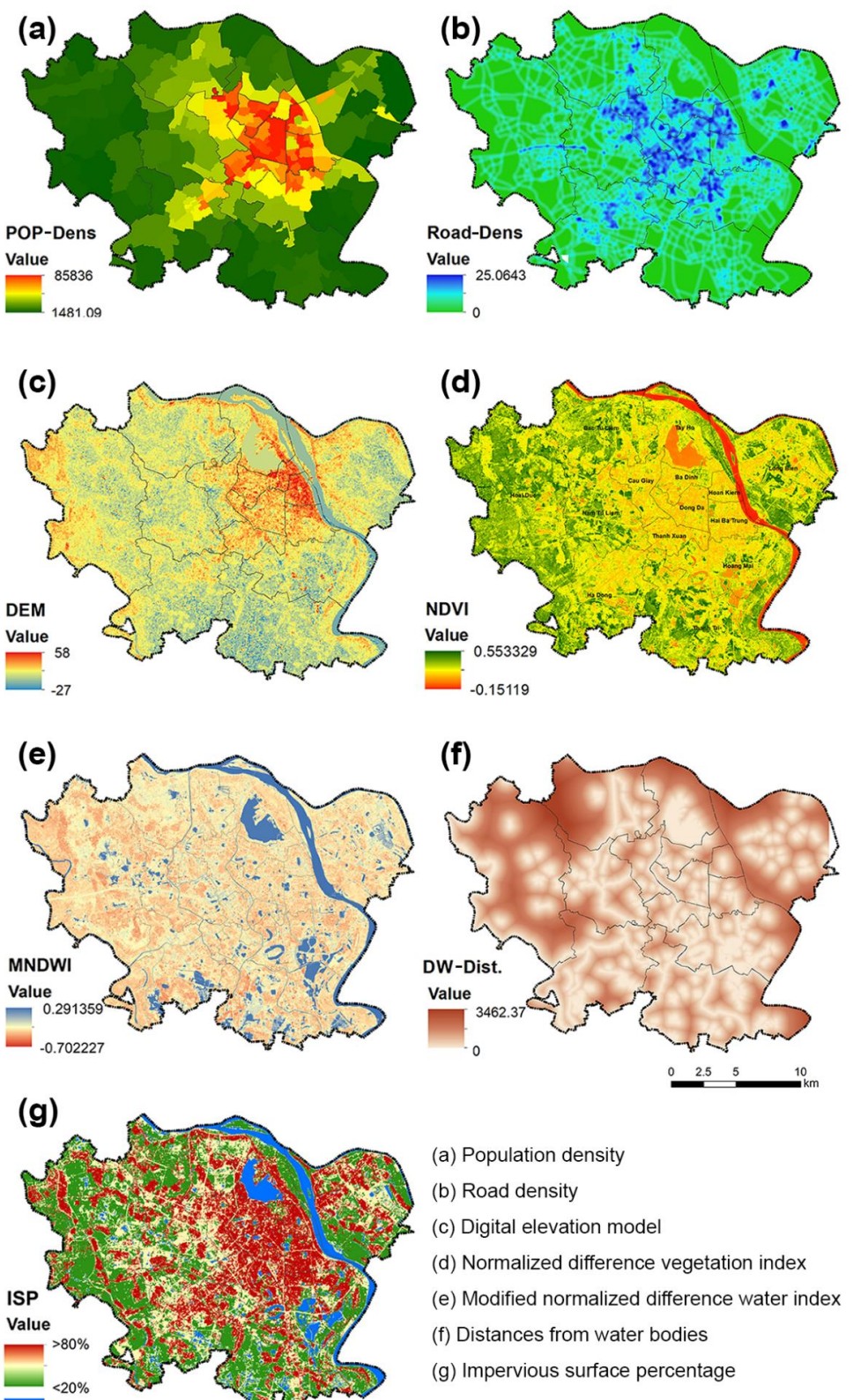

**Figure 3.** Spatial distribution of factors in the study area. Abbreviations: POP-Dens, population density; Road-Dens, road density; DEM, digital elevation model; NDVI, normalized difference vegetation index; MNDWI, modified normalized difference water index; DW-Dist, distances from water bodies; ISP, impervious surface percentage.

Previously, Hanoi was located in a landscape with a rich lake and river network, but swift urbanization has caused many natural lakes, ponds, and ditches to be filled and converted into residential areas or industrial parks. As the underground drainage system is outdated, urban waterlogging is becoming more serious. Therefore, we took the characteristics of the distances from the waterlogging spots to the rivers and lakes in Hanoi into consideration in order to explore the relationships between the distances from water bodies (WD Dist) and urban waterlogging risks. The attribute values of the WD Dist map (Figure 3f) were extracted from the MNDWI map (Figure 3e). The modified normalized difference water index (MNDWI) provided important quantitative information about the water bodies of areas, which, by the use of the ArcGIS tool, can help to generate maps for water bodies in a study area [114].

In addition, the normalized difference vegetation index (NDVI) provides important quantitative information for measuring the status of vegetation coverage on the ground and reflects the important ecological effects of vegetation on the urban ecological system [115,116]. High-density vegetation coverage has a great potential to disrupt flow, provide numerous ecosystem functions and efficiency in stormwater retention, and reduce urban waterlogging risks (Figure 3d) [117].

Rapid urbanization is accelerating the transformation of urban land use, usually by reducing natural land and increasing the impervious surface areas, and is affecting the social economy and the environment in Hanoi [112]. Much research shows that an impervious surface is one of the most important factors for urban rainstorm waterlogging. As impervious surface information is very easily obtained through remote sensing images for urban area [117–122], we used impervious surface percentage (ISP) as one of the explanatory variables in our regression model (Figure 3g).

*3.4. Integrated Frameworks*

The spatial framework includes three modules for the spatial analysis of the risks of urban waterlogging: data processing, attribute extraction, and regression modeling (Figure 4). In the data-processing module, information about urban waterlogging spots was digitalized by the ArcGIS software to construct the spatial distributions of the waterlogging spots. We adopted 500 m as the urban waterlogging risk radius, and the spatial buffer technique of ArcGIS was employed to construct a waterlogging risk area vector, which is considered to be a data vector for integrating the data of the layers. KDE was used to calculate the waterlogging spot density (Figure 6) and is considered to be the dependent variable in our regression model. Moreover, HSA was conducted (Figure 7) to evaluate the relationships of the waterlogging spots and the results from the KDE and HSA tools we predicted for the six spatial factors (Figure 4), which are considered to be the explanatory variables in our regression model. Finally, all spatial factor datasets were converted to a raster with a cell size value of $30 \times 30$ and were clipped by the boundaries of the study area (Figure 3). In the attribute's extraction module, ZST and EMVP were conducted to calculate statistics for the values of the one dependent variable raster and the six explanatory variable rasters in order to form a waterlogging risk dataset. In the regression modeling module, we considered the results from OLS to check for the relationships between the dependent and explanatory variables, as well as to discover all of the key variables that would explain our model. If the results show that we have a properly specified model (a trustworthy model) that is statistically significant, then we would be able to improve the model's results by moving to GWR to compensate for the disadvantage of OLS in the prediction of urban waterlogging risks.

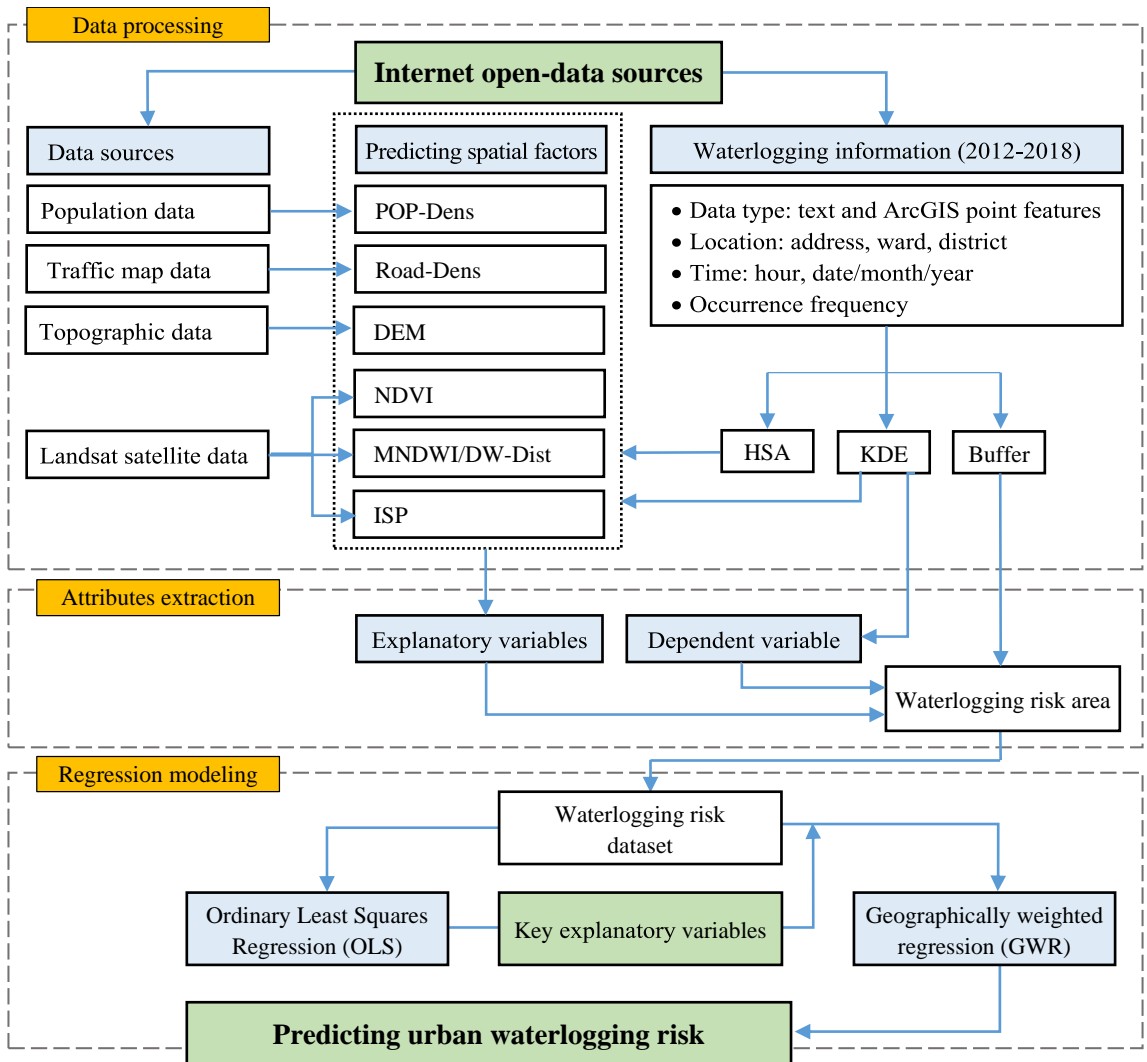

**Figure 4.** Integrated Framework of Regression model. Abbreviations: HSA, Hot Spot Analysis; KDE, Kernel Density Estimation.

## 4. Results and Discussion

### 4.1. Waterlogging Frequency and Spatial Distribution Characteristics of Waterlogging Spots

Figure 5 shows the distribution characteristics of the waterlogging spots caused by rainstorms in Hanoi during the period 2012–2018. The range of the waterlogging occurrence frequency was 1–11 times, most of which was 1–2 times (57.4%), followed by 3–4 times (20.3%), 5–7 times (10.8%), and eight times or more (11.5%) (Figure 5). The majority of the high-frequency waterlogging spots were located in the inner districts of Hanoi, and the frequencies of the suburban districts were lower. Therefore, we preliminarily decided that the waterlogging occurrence frequencies had a certain dependence on space.

Using 148 waterlogging spots, the KDE calculated the waterlogging spot densities, the highest of which was concentrated in the inner city of Hanoi and spread to the northwest, west, southeast, south, and northeast of the city, with a tendency to continue expanding into the surrounding ring roads, radial axis roads, and traffic intersections (Figure 6). The spatial autocorrelation tool was also run on the value of the dependent variable after attribute extraction from the results of the KDE, which produced a Moran's index of 0.18, a $z$-score of 11.8, and a $p$-value of 0.00, indicating that the occurrences of

waterlogging events in the study area were not randomly distributed and were spread to adjacent areas in a spatial form of clustering.

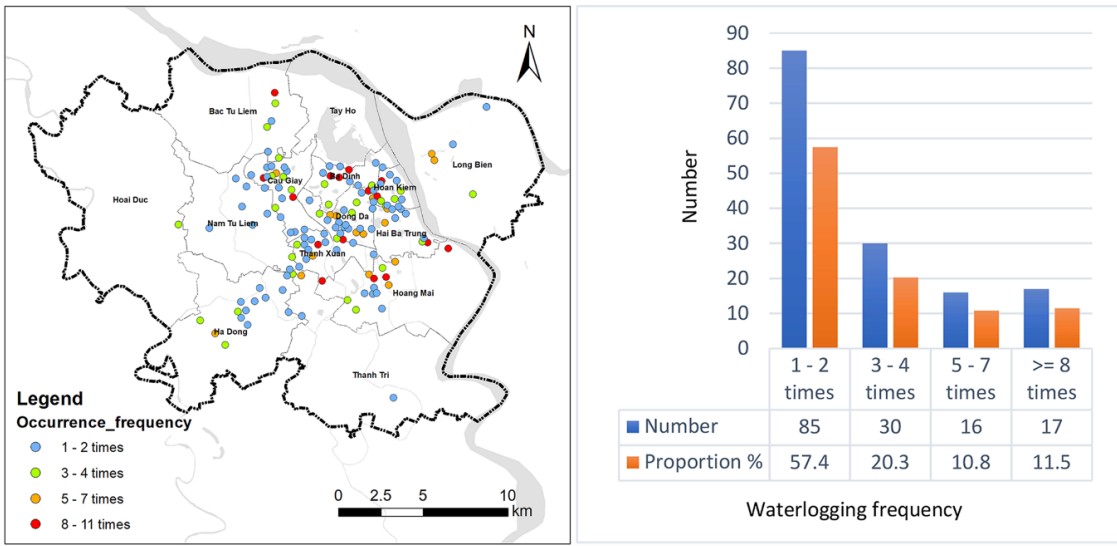

**Figure 5.** Waterlogging occurrence frequencies caused by rainstorms in the period 2012–2018.

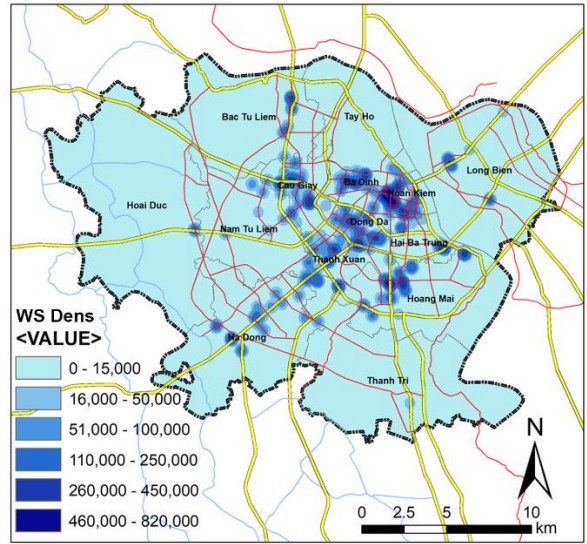

**Figure 6.** Waterlogging spot density (WS-Dens) map.

Next, HSA was conducted to evaluate the existence and relationships of waterlogging events. Areas with high or very low waterlogging risks are shown in red (hot spot) and blue (cold spot), respectively. Dong Da District has many hot spots, with a spatial distribution that is densely distributed in the city center (Figure 7). Case studies of the spatial factors of Dong Da District have revealed a high population density (Figure 8b), high road density (Figure 8c), medium urban surface elevation (Figure 8d), low-density vegetation coverage (Figure 8e), low modified normalized difference water index (Figure 8f), and high impervious surface percentage (Figure 8g).

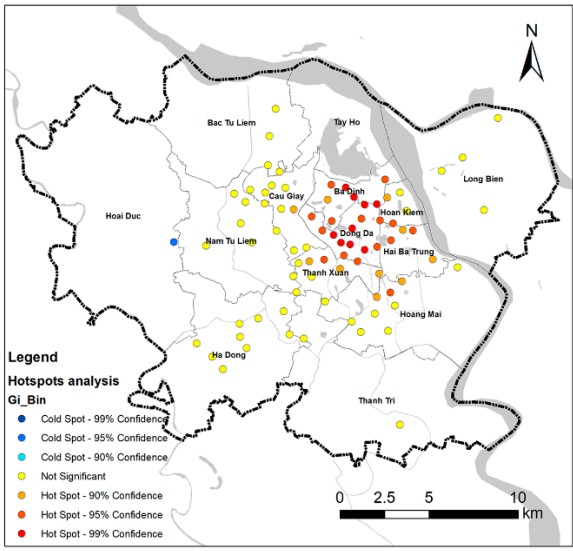

**Figure 7.** Hotspot analysis map.

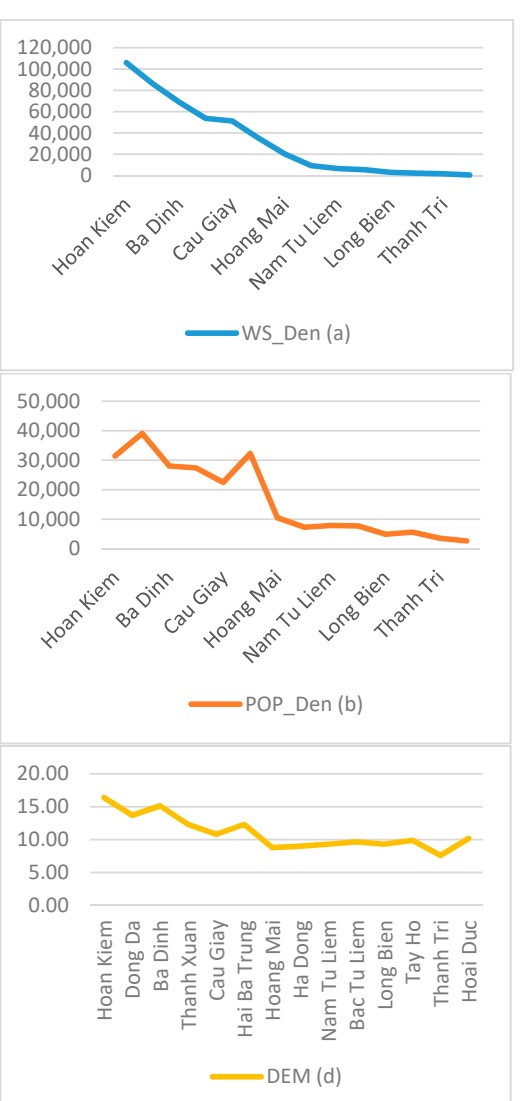

(a) Waterlogging spot density;
(b) Population density;
(c) Road density;
(d) Digital elevation model;
(e) Normalized difference vegetation index;
(f) Modified normalized difference water index;
(g) Impervious surface percentage.

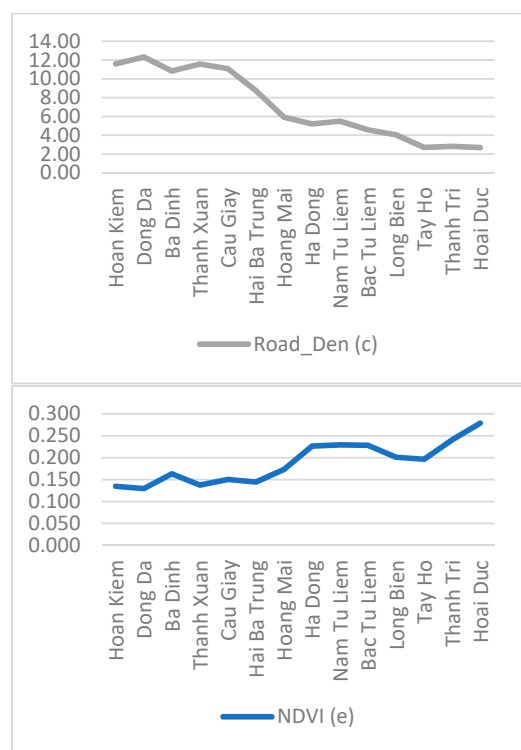

**Figure 8.** *Cont.*

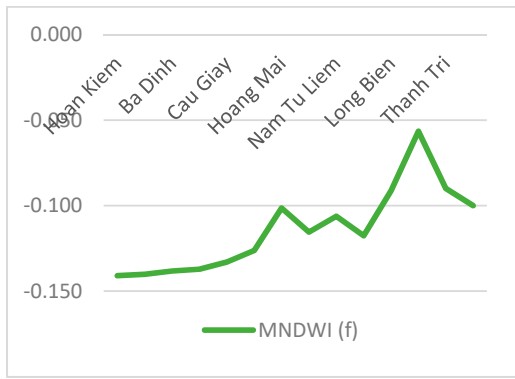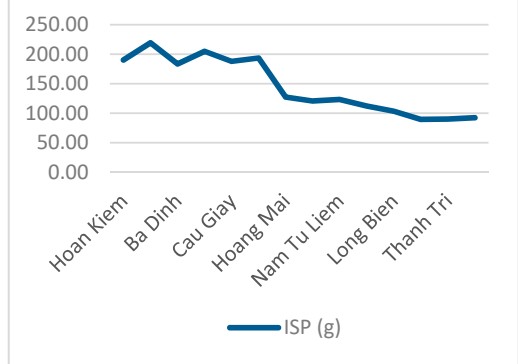

**Figure 8.** The median values of the spatial factors for each district.

Therefore, we decided preliminarily that the urban waterlogging risk in the study area had a certain relevance to these spatial factors.

From the above results, we were able to draw several obvious conclusions. First, the waterlogging events caused by rainstorms in Hanoi during the period 2012–2018 were densely distributed in the city center and had ab tendency to continue to expand into suburban areas at locations surrounding the ring and radial axis roads, traffic intersections, and densely populated areas, similar to the urban surface expansion affected by urbanization. Second, the waterlogging occurrence frequency has a certain dependence on space, which means that the locations of the urban waterlogging events were not randomly distributed. Third, the six spatial factors—POP-Dens, Road-Dens, DW-Dist, NDVI, ISP, and DEM—of urban surfaces were considered as the explanatory variables of the regression model to explain the urban waterlogging risks.

*4.2. OLS Analysis*

According to the above analysis, the locations of the urban waterlogging spots are not randomly distributed. Thus, OLS analysis, which is based on the assumption of random distribution, would not be suitable for predicting urban waterlogging risks. Therefore, the purpose of OLS analysis would be to check for relationships between the dependent and explanatory variables, in order to find all of the key variables that explain the observed urban waterlogging risks and to determine if the regression model is well specified.

4.2.1. Finding Key Explanatory Variables

Using the above six spatial variables, we iterated many univariate and multivariate OLS models, and then we selected two typical models. In Model 1, we tested the hypothesis that all of the six spatial factors of urban surfaces were important explanatory variables. The results show that, except for DEM ($p > 0.01$), the coefficients of the factors are significant ($p < 0.01$ *) (Table 2), indicating that the DEM variable could not help the model. In Model 2, we conducted an OLS analysis of the five significant spatial factors to check the results from the adjusted $R^2$ value, which is a confidence level (0–1) that is explained by the model through explanatory variables, to determine if the DEM variable should be removed. Table 2 shows that the value (0.659411) of the adjusted $R^2$ (Adjusted-$R^2$) of Model 1 is not much higher (0.654604) for this new model, indicating that the DEM variable does not improve the performance of the model. Hence, the mean DEM is not an important explanatory factor in this study area, so we should remove this variable. This result agrees with a previous study by [123] in the Netherlands, which concluded that pluvial floods are not linked to urban topography. In fact, the terrain of the study area is relatively flat and the digital elevation difference is low, so this result is completely suitable. Therefore, in this study, only POP-Dens, Road-Dens, DW-Dist, ISP, and NDVI were determined as the key explanatory variables for predicting urban waterlogging risk.

**Table 2.** Comparing relevant parameters of both ordinary least squares (OLS) models.

| Model | Explanatory Variables | Coefficient | Coefficient Statistical Significance (*p*-Value) | Adjusted-R$^2$ |
|---|---|---|---|---|
| Model 1 | POP-Dens | $2.094 \times 10^0$ | $p < 0.01$ * | |
| | Road-Dens | $1.931 \times 10^4$ | $p < 0.01$ * | |
| | NDVI | $-8.936 \times 10^5$ | $p < 0.01$ * | 0.659411 |
| | DW-Dist | $6.941 \times 10^1$ | $p < 0.01$ * | |
| | DEM | $-3.371 \times 10^3$ | $p > 0.01$ | |
| | ISP | $1.589 \times 10^5$ | $p < 0.01$ * | |
| Model 2 | POP-Dens | $1.514 \times 10^0$ | $p < 0.01$ * | |
| | Road-Dens | $2.067 \times 10^4$ | $p < 0.01$ * | |
| | NDVI | $-8.742 \times 10^5$ | $p < 0.01$ * | 0.654604 |
| | DW-Dist | $6.546 \times 10^1$ | $p < 0.01$ * | |
| | ISP | $1.451 \times 10^5$ | $p < 0.01$ * | |

Note: Asterisk (*) indicates a coefficient is statistically significant ($p < 0.01$).

### 4.2.2. Checking Relevant Parameters of OLS Model

We had to check some issues before we were satisfied that we had a properly specified model. First, the OLS's default output is a map of the model's residuals. The Spatial Autocorrelation tool was run on the values of the residuals in the OLS default output class. The results produced a Moran's index of 0.013, a *z*-score of 1.212, and a *p*-value of 0.225, indicating that the model residuals were randomly distributed and that we had a properly specified OLS model. Second, in the Coefficients column (Table 3), we checked the signs (+/−) before the coefficients of the explanatory variables.

**Table 3.** Relevant parameters of OLS Model 2.

| Explanatory Variables | Coefficient | Probability | Robust_Pr | VIF | Adjusted-R$^2$ | AICc | BP |
|---|---|---|---|---|---|---|---|
| Intercept | $-3.411 \times 10^5$ | $p < 0.01$ * | $p < 0.01$ * | - | | | |
| POP-Dens | $1.514 \times 10^0$ | $p < 0.01$ * | $p < 0.01$ * | 2.61 | | | |
| Road-Dens | $2.067 \times 10^4$ | $p < 0.01$ * | $p < 0.01$ * | 2.21 | 0.654604 | 3598.435 | $p < 0.01$ * |
| NDVI | $-8.742 \times 10^5$ | $p < 0.01$ * | $p < 0.01$ * | 1.83 | | | |
| DW-Dist | $6.546 \times 10^1$ | $p < 0.01$ * | $p < 0.01$ * | 1.34 | | | |
| ISP | $1.451 \times 10^5$ | $p < 0.01$ * | $p < 0.01$ * | 3.11 | | | |

Note: Asterisk (*) indicates a coefficient is statistically significant ($p < 0.01$).

The results show that waterlogging frequency was positively related to four spatial factors—POP-Dens, Roads-Dens, DW-Dist, and ISP—but negatively related to NDVI, indicating that if the median values of POP-Dens, Roads-Dens, DW-Dist, and ISP in the area increased, but that of NDVI decreased, then the waterlogging frequency would increase as well. Next, we checked the Probability and Robust_Pr columns to see if all of the coefficients were statistically significant ($p < 0.01$*), indicating that these five variables helped to explain the urban waterlogging risk with a confidence level of 65.5% (adjusted-R$^2$ column). The value of Akaike's Information Criterion (AICc) was also used to measure the model's performance. When we had a GWR candidate model, we were able to determine which model was the best by looking for the lowest AICc value (Table 4). In the Variance Inflation Factor (VIF) column, the values of all variables less than 7.5 showed no redundancy among the explanatory variables. Finally, the Koenker Statistic (BP) test was one OLS diagnostic that was very important. This test is statistically significant ($p < 0.01$) if one variable may be an important predictor of urban waterlogging risk for this location but possibly a weak predictor for other locations, indicating that the relationships between all the explanatory and dependent variables are non-stationary (or not spatially random or heteroskedastic). This result would be completely consistent with our second conclusion, mentioned in Section 4.1. The above test results indicate that modeling for predicting

urban waterlogging risk in the study area as a function of POP-Dens, Roads-Dens, DW-Dist, ISP, and NDVI would produce a proper model. Next, we checked the GWR model to improve the efficiency of the predictive model.

**Table 4.** Relevant parameters of the geographically weighted regression (GWR) model.

| Research Period | Spatial Autocorrelation Report | | | GWR Model | |
| --- | --- | --- | --- | --- | --- |
| | | | | Adjusted-$R^2$ | AICc |
| 2012–2018 | Moran's I = 0.006 | $z$ = 0.844 | $p$ = 0.398 | 0.676 | 3616.753 |

*4.3. GWR Analysis*

4.3.1. Performance of GWR Model

On the basis of the waterlogging risk dataset and key explanatory variables of the OLS Model 2, GWR was conducted to predict the urban waterlogging risk. The GWR's default output is a map of the model's residuals. In ArcGIS, the Spatial Autocorrelation tool was run on the values of the residuals in the GWR default output class. The results produced a given $z$-score of 0.844 (Table 4), indicating that the model residuals were randomly distributed and that we had a properly specified GWR model. In addition, the results of the GWR analysis show that the adjusted $R^2$ value is higher (67.6%) for the GWR model than for the above OLS model (65.5%), while the AICc value is lower for the GWR model, with a decrease of more than 18.4 points (OLS is 3598.4; GWR is 3616.8), which indicates a real improvement in the model's performance (Tables 3 and 4). The adjusted $R^2$ for the GWR model is 0.676, indicating that the five spatial factors—POP-Dens, Road-Dens, DW-Dist, ISP, and NDVI—of urban surfaces could explain the urban waterlogging risks with a confidence level of 67.6%. The adjusted $R^2$ is low due to the lack of other spatial factors, such as land subsidence, underground drainage systems, pump stations, and soil water retention. These factors were not considered in this study, owing to the fact that spatially fine-scale data are currently unavailable.

4.3.2. Regression Coefficient of GWR Model

The regression coefficients of the GWR model are shown in Figure 9. The signs (+/−) before the GWR coefficients of the explanatory variables are the same as those of the OLS model. First, the coefficients of population density are positively related to waterlogging frequency. Therefore, the increase in population density in an area is related to a higher waterlogging risk. The majority of the stronger influence of population density on waterlogging frequency was located in the southwest of Ba Dinh, west of Dong Da, north of Thanh Xuan, and east of Cau Giay (Figure 9a).

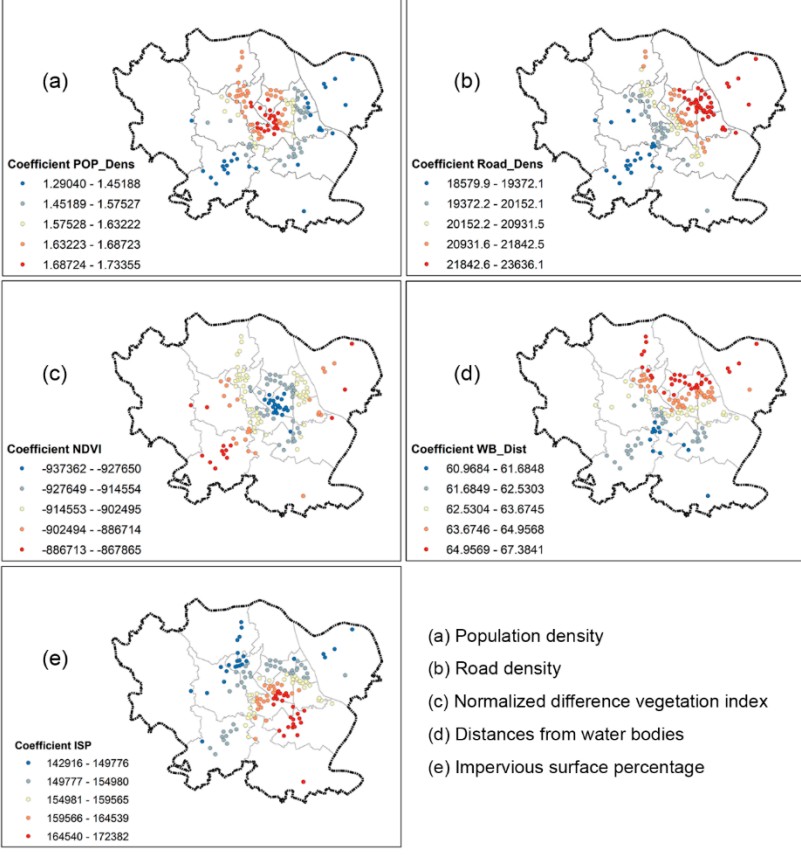

**Figure 9.** Regression coefficients of the GWR model.

As we know, rapid urbanization caused by population growth is accelerating the expansion of impervious surfaces, in addition to outdated underground drainage systems. Together, these two factors have probably played a critical role in increasing the waterlogging frequency in Hanoi. Second, the coefficients of road density are also positively related to waterlogging frequency, indicating that if the median value of the road density in the area increases, then the waterlogging frequency would also increase. A stronger influence of road density on waterlogging frequency was located in the Ba Dinh, Hoan Kiem, Dong Da, and Hai Ba Trung Districts, as well as at the intersections with the ring or radial axis roads in the Long Bien, Hoang Mai, and Bac Tu Liem Districts (Figure 9b). This result agrees with the conclusions of [62], who considered that an increase in road density may increase the waterlogging frequency. Third, the coefficients of NDVI are negatively related to waterlogging frequency, implying that if the median value of NDVI decreases, then the waterlogging frequency would increase. The influence of NDVI was relatively weaker in the Dong Da, Ba Dinh, and Hoang Mai Districts (Figure 9c).

This result agrees with a large number of studies [124–126], which concluded that vegetation can reduce flood risk because it offers a greater infiltration capacity and evapotranspiration potential. Fourth, the coefficients of the distances from water bodies are also positively related to the waterlogging frequency, and their influence was relatively stronger in Ba Dinh, Hoan Kiem, Dong Da, north of Cau Giay, and east of Bac Tu Liem (Figure 9d), indicating that areas far from natural drainage systems have higher waterlogging frequencies. Because of the swift rural–urban transformation, increases in the amount of built-up land probably result in the filling of lakes and rivers. The disappearance of water bodies has also been posited as one of the important factors leading to urban waterlogging in Hanoi. Finally, the coefficients of impervious surface percentage are also positively related to waterlogging frequency, indicating that an increase in impervious surface in an area is related to higher waterlogging risk. The stronger influence of impervious surface percentage on waterlogging frequency was located

in the Dong Da, Thanh Xuan and Hoang Mai Districts (Figure 9e). This result agrees with a number of studies [12,26] which concluded that an increase in the impervious surface area may be one of the most common causes of urban waterlogging. The rapid urbanization process in Hanoi has led, and will continue to lead, to the rapid growth of impervious surface areas. Impervious surfaces reduce the infiltration capacity of the urban surface to rainwater, resulting in an increase in the volume of surface water, and then aggravate urban waterlogging risks.

## 5. Conclusions

In conducting a case study in Hanoi with data from the years 2012 to 2018 using the ArcGIS platform, we applied a regression model with open-data sources from the internet, to predict urban waterlogging risks. As a regression model, which is a method that provides powerful and reliable statistics for examining and estimating linear relationships, KDE was used to calculate the values of the dependent variable. Six spatial factors as explanatory variables were predicted to construct the regression model. Because the locations of the urban waterlogging spots are not randomly distributed, OLS was not appropriate for predicting urban waterlogging risks. However, in this study, OLS was used to check the key explanatory variables for the regression model and to determine if the model was well specified. The results indicated that the modeling for predicting urban waterlogging risks in the study area as a function of POP-Dens, Roads-Dens, DW-Dist, ISP, and NDVI could produce a proper model. These five spatial factors have a huge impact on urban waterlogging risk, although this impact is different in different areas. In addition to OLS, GWR was utilized to predict the urban waterlogging risk, and achieved a good modeling effect for predicting urban waterlogging risks, with a confidence level of 67.6%.

The novelties of this study lie in two main aspects. First, with internet open-data sources, a regression model was applied to predict urban waterlogging risk at a high confidence level. Second, the methods of attribute extraction, as well as the selection and checking of the variables, are considered key to successful predictions of urban waterlogging risks.

The regression model can be applied to other cities in the future to predict urban waterlogging risks. On the basis of the natural–social conditions and the infrastructures of study areas, models will be developed by the consideration of additional factors (precipitation, topographical features, underground drainage systems, pump stations, soil water retention, etc.) to explain urban waterlogging risks with higher confidence levels.

**Author Contributions:** D.T. proposed the basic idea, designed the approaches involved in this study, collected and processed the data, performed the analysis and wrote the paper. D.X. proposed the original concept, reviewed the research findings. V.D. collected the data. A.A.Q.A. provided the useful suggestions on designing the software. All authors have read and agreed to the published version of the manuscript.

**Funding:** This research was funded by the Chinese Scholarship Council and The APC was funded by Northeast Forestry University.

**Acknowledgments:** This study was conducted in the College of Landscape Architecture at Northeast Forestry University, Harbin, Heilongjiang, China which was jointly finished by Northeast Forestry University and Vietnam National University of Forestry. The authors would like to thank the reviewers for their valuable remarks and comments.

**Conflicts of Interest:** The authors declare no conflict of interest.

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
