# Peer review of "Predicting Urban Waterlogging Risks by Regression Models and Internet Open-Data Sources"

_water, doi:10.3390/w12030879_

Round 1

Reviewer 1 Report

The methodology described in the paper could be improved taking into account the hydrological characteristics of the events that generated the waterlogging (for example the rainfall height etc.). These characteristics are key factors in the flood risk assessment and should be included among the analyzed factors. For which, I suggest accepting the paper with major revisions. Finally, English needs to be properly revised.

Author Response

Dear Reviewer 1,

First of all, we would like to thank you for the time and effort you invested in reading and reviewing this manuscript. We appreciate your contribitions to our work!

We have implemented major changes to the original version of the manuscript, and hope this revised manuscript has been enough to consider publishing in Water.

For detailed replies to your specific comments, please refer to the attached document.

Thank you for your consideration!

Reviewer 2 Report

The first two paragraphs of the introduction do not provide a strong context, nor adequately set the need for this study. Some of the statements are contradictory to what comes later in the introduction, for example Line 45 says there is a current lack of an urban database and the Line 81 says that open urban data are very complete.

Introduction

Line 37-“social economy” do you causing economic losses or do you mean the social economy as in enterprises that use market mechanisms to pursue economic and social objectives?

Line 39-Do the 35% and 56% indicate the percentage of the population living in urban areas or is it the rate of urbanization?

Lines 39-41-This sentence is redundant and should be removed.

Lines 42-44-Are there studies to support the claims of “serious urban waterlogging”, provide context to what serious means, i.e. frequency of flooding, damage associated with such flooding.

Line 45-An urban database for what type of data?

Line 47-What do you mean by the urban imbalance caused by urbanization? Do you mean that imbalance between rural land and urban land?

Lines 57-59-Do you mean that in comparison to extensive literature on risk assessment using simulation models, there is a lack of information or studies examining the factors controlling the spatial-temporal variability in the extent and impact of urban waterlogging?

Line 84-you state that many studies have “achieved significant achievements”, what do you mean by achievements? Models results were validated and showed strong predictive abilities? The application of open data to hydrological analyses was used for management purposes? The citation you provided refers to one case study only.

Line 85-“From our study, ….drawn conclusions”. This really is just a summary of your literature review findings. Your second point could be incorporated into the paragraph above or alternatively, this paragraph could go earlier in the introduction and serve as more of an outline of the argument you are making for the use of open source data for geo-statistical analysis of urban waterlogging.

Study Area

Line 109-since monsoonal rains are a significant contributor to waterlogging, the annual distribution of precipitation is needed, as well as mean rainfall intensity values.

Line 111-where is old Hanoi located on Figure 1?

Line 117-what is meant by a “rich” lake and river network?

Line 118-what is the extent of impervious cover? In comparison to other land covers such as surface water, rural, forested, crop?

Line 118-what is meant by “swift”? At what rate has land cover changed from rural to urban? How has urban coverage increased relative to rural? Have you analyzed land cover change using ArcGIS?

Line 119-a citation is needed to support the statement regarding the adverse impacts to dikes, pumping stations, etc.

Lines 118-128-this discussion should be part of the introduction because it is setting the need for the study in context of increasing events and threats to public safety.

Data Sources

Line 132-By “waterlogging spots” do you mean areas prone to flooding/waterlogging?

Line 135-How did you define the extent of “waterlogging spots”, i.e. how was the spatial extent/boundary defined?

Line 136-what type of classification was used and what were the land cover categories for the Landsat data?

Figure 2- for each year with more than one storm, how many areas corresponded to each storm?

Table 1-it is unclear how you used each of these layers. Does this table have value? It could be combined with Figure 4.

Waterlogging Risk Areas

Lines 167-170-What do you mean by location, time and occurrence frequency? How did you determine the extent of the waterlogged area? Was this done using ArcGIS? It’s not clear what you mean by frequency. In 2012 you list only one rainstorm. How did you digitize boundaries, not using Arc tools to determine areas prone to flooding?

Attribute Extraction

Line 217-why were the distance characteristics of waterlogging spots to rivers and lakes determined? Explain here rather than Line 251.

Explanatory Variables

Line 229-NDVI sourced from NOAA-AVHRR, this source isn’t in Table 1.

Line 244-Replace “altitude” with elevation.

Line 246-Figure 3 is more appropriate for the study area discussion as opposed to a results discussion. These maps are showing extent of cover for each category. It would have been more valuable to show the conversion of non-urban land cover to urban land cover over the 2012-2018 period.

Line 255-what hydrological characteristics are you referring to specifically?

Figure 5-What does number refer in the bar graph? It is very difficult to determine what rainfall events correspond to what waterlogging areas and over what time period you are counting frequency of occurrence.

Author Response

Dear Reviewer 2,

First of all, we would like to thank you for the time and effort you invested in reading and reviewing this manuscript. We appreciate your contribitions to our work!

Considering your valuable comments, we have implemented major changes to the original version of the manuscript. The most relevant changes of this major revision are the more explicit focus on concepts and terminology, the restructure the content of the introduction, the additional literature review, the improvement of the language, and the addition of some new references. We are convinced that your helpful comments have helped us to improve the quality of our manuscript, and hope this revised manuscript has been enough to consider publishing in Water.

For detailed replies to your specific comments, please refer to the attached document.

Thank you for your consideration!

Reviewer 3 Report

The paper is interesting and well written, the author can improve the language and added some newest reference.

Author Response

Dear Reviewer 3,

First of all, we would like to thank you for the time and effort you invested in reading and reviewing this manuscript. We appreciate your contribitions to our work!

Considering your valuable comments, we have implemented major changes to the original version of the manuscript. The most relevant changes of this major revision are the more explicit focus on concepts and terminology, the restructure the content of the introduction, the additional literature review, the improvement of the language, and the addition of some new references. We are convinced that your helpful comments have helped us to improve the quality of our manuscript, and hope this revised manuscript has been enough to consider publishing in Water.

For detailed replies to your specific comments, please refer to the attached document.

Thank you for your consideration!

Round 2

Reviewer 1 Report

The manuscript has been significantly improved. The answers provided to my comments are exhaustive. Therefore, the manuscript can be accepted in present form.

Reviewer 2 Report

Thank for the changes made, particularly related to the introduction, your paper's focus and contribution is much clearer.